# OpenReview forum: "What Makes a Strong Model? A Unified Spectral Analysis of Knowledge Transfer over High-dimensional Linear Regression"
_ICML.cc/2026/Conference — ICML 2026 regular_

### Official Review · Reviewer_rAs7 · 2026-02-18

**Soundness:** 3
**Presentation:** 3
**Significance:** 3
**Originality:** 3
**Overall Recommendation:** 4
**Confidence:** 4

**Summary:**

This paper examines when knowledge transfer is actually helpful. It uses a high-dimensional linear regression model with spectral assumptions. The main result breaks down the transfer excess risk into three parts: (I) teacher error, (II) student optimization error, and (III) an alignment bias term. These relate to an “effective dimension” cutoff $k^*$. The paper also identifies conditions where distilation is more sample-efficient than direct learning, including a heavy-tailed miss-specified case where the efficiency ratio grows with model “strength”.

**Compliance With Llm Reviewing Policy:**

Affirmed.

**Key Questions For Authors:**

see the weaknesses above

**Limitations:**

yes

**Strengths And Weaknesses:**

Strengths

1. The risk decomposition is helpful because it makes the common idea that “the teacher helps the student” more concrete, especially by clearly separating optimization error from alignment bias.

2.  The geometric consistency condition $\Pi_S\Pi_T^{\perp}w^*=0$ nicely formalizes when transfer will not harm performance in the long run.

3. The heavy-tail result provides an easy-to-understand scaling law showing that distillation can be much more efficeint than direct learning.



Weaknesses

1. The analysis is done entirely within a linear regression and spectral-decay framework, with conclusion based on $\alpha_T,\alpha_S$ and eigenvalue decay (for example, $\lambda_k\propto k^{-\alpha_S}$). While this works for theory, the paper sometimes sounds like it’s explaining “what makes a strong model” in general, eventhough the scope is quite specific.

2. Much depends on alignment, but the paper does not really show how to measure it in practice. Lemma 2 states that transfer matches direct learning if and only if $\Pi_S\Pi_T^{\perp}w^\star=0$; otherwise, there is an irreducable bias $\frac12|\Pi_S\Pi_T^{\perp}w^\star|^2>0$. This is a clear and strong statement, but in real applications, we need a measurable proxy or at least a discussion on how to estimate $|\Pi_S\Pi_T^{\perp}w^\star|$ from data. Without this, it’s hard to know when transfer might cause harm.


3. There are some minor but annoying inconsistencies in the math and notation.
  * The definition of the effective dimention $k^\star$ seems to change throughout the paper. In Theorem 1, it’s $k^\star=\max{k:\lambda_{k,S}>\delta \log^2 n/(4\gamma_0’ n)}$ (with $\log_2 n$ also mentioned), but later Theorem 9 uses $k^\star=\max{k:\lambda_{k,S}> 2\ln n,\log_2 n/(\gamma_0’ n)}$. These differ in constants, log factors, and even log bases. It’s unclear if this is intentional or a loose restatement, so clarification is needed.
  * Theorem 1’s bound include many terms with projection operators and moment matrix, but some objects like $P_{n,T2S}$ and $P^{\mathrm{var}}_{n,T2S}$ appear without a clear introduction. Although they’re defined later in the theorem, the presentation feels dense and hard for first-time readers.


4. “Strength” parameterization is a bit counterintuitive and needs more explanation. The paper states “if the Student is strictly weaker ($\alpha_S>\alpha_T$) … DERN diverges” in Theorem 3. Since larger $\alpha$ typically means faster spectral decay, calling it “weaker” is plausible, but this is not obvious and I had to stop and check what direction the inequality should go. I think a short intuition parageraph early would help a lot (like one sentence: larger $\alpha$ means fewer effective dimensions learned at a given sample size).


5. Some claims feel under-supported by experiments (at least from what is shown). The theory predicts different regimes: transfer helps by replacing the student’s approximation bias term $|\Pi_S^{\perp}w^\star|^2$ with the teacher’s smaller bias $|\Pi_T^{\perp}w^\star|^2$ (the “effective noise” argument), and also predicts a strong scaling advantage in heavy-tailed tasks where $\beta_i^2=\Theta(i^{\beta})$. But I didn’t see (in the parts provided) a careful empirical check that isolates these pieces, for example measuring the alignment bias proxy, or showing that performance differences track the predicted cutoff $k^\star$. Right now it’s mostly theoretical narrative, with limited “sanity checks”.

6. The paper gives asymptatic statements like $\lim_{n\to\infty}\mathrm{DER}N>1$ and $\lim{n\to\infty}\mathrm{DER}_N=\Omega(\tilde e,N^{\kappa})$, plus a $1-\mathrm{PGR}$ scaling (later derivation shows $1-\mathrm{PGR}\asymp N^{-,\frac{2\alpha_S}{\alpha_T(2\alpha_S+1)}}$). This is nice, but I’d like a clearer “interpretation layer”: what is the recommended finite-$N,n$ way to estimate DER from runs, and how sensitive is it to early stopping schedule and $\delta$ choice? Otherwise it stays as asymptatic math that is hard to apply.

---

> ### Author Rebuttal · Authors · 2026-03-31
>
> W1: Linear regression and spectral decay framework.
>
> A1: The key phenomena we identify (spectral horizon expansion, $\delta$-damping) arise from the implicit regularization of SGD, for which linear regression provides rigorous analysis.
> This is not merely a toy setting: recent empirical work (e.g., Malladi et al., 2023) shows that fine-tuning dynamics of modern networks (including LLMs) are well approximated by kernel (i.e. linear) regression. As for the polynomial decay $\lambda_k\propto k^{-\alpha}$, our core results (Lemma 1, Theorems 1) hold for **arbitrary spectra**.
> The polynomial form is invoked in Theorems 3 and 5 to derive explicit rates, following standard practice in spectral learning theory.
>
> W2: Measurable proxy for alignment bias.
>
> A2: To test whether $\|\Pi_S\Pi_T^\perp w^\star\|\approx 0$ holds in practice, we design a simple probe.Given a trained teacher $f_T$ and a pretrained student feature map $\phi_S$, we fit two regressions from $\phi_S(x_i)$: one predicting the teacher output $f_T(x_i)$ (coefficient matrix $W_p$) and one predicting the residual $r_i=y_i-f_T(x_i)$ (coefficient matrix $W_r$). If the student features cannot capture what the teacher misses, $\|W_r\|_F^2$ should be negligible relative to $\|W_p\|_F^2$. Indeed, on ImageNet with ViT-B-16 (teacher) / ResNet-34 (student), we find $\|W_r\|_F^2/(\|W_r\|_F^2+\|W_p\|_F^2)=0.021/(0.021+0.26)=0.073$, confirming that $\Pi_S\Pi_T^\perp w^\star\approx 0$ is a reasonable assumption.
>
> W3.1: Varying definitions of $k^\star$ across theorems.
>
> A3.1: The variation is intentional. Both $k^\star_\nu$ and $k^\star$ represent the same physical quantity—the effective dimension—but serve different technical roles: $k^\star_\nu$ in Lemma 1 uses a $2\ln N$ factor to guarantee $\mathcal{O}(1/N^2)$ bias decay in the learned head, while $k^\star$ in Theorem 1 uses a $\delta/4$ threshold to delineate the boundary where $\delta$-damping emerges. The differing log factors and constants are artifacts of the respective proof techniques, not a change in meaning. We clarify this in the revision.
>
> W3.2: Readability of Theorem 1.
>
> A3.2: We thank the reviewer for this presentation feedback. We will adjust the presentation accordingly in the revision.
>
> W4: Clarify why larger $\alpha$ means "weaker'' and add early intuition.
>
> A4: We thank the reviewer for this suggestion and insightful question. The connection operates through two complementary angles.
> **(Optimization angle)** Under $\lambda_k \asymp k^{-\alpha}$, the effective dimension satisfies $k^\star \propto N^{1/\alpha}$: a larger $\alpha$ means eigenvalues decay faster, so at a given sample size $N$ fewer dimensions cross the learning threshold $\lambda_k \gamma N \gtrsim 1$. Hence larger $\alpha$ $\Rightarrow$ fewer features learned $\Rightarrow$ weaker model.
> **(RKHS capacity angle)** As shown in Remark 3 (Appendix D.1), a function $f = \sum_k c_k \psi_k$ belongs to $\mathcal{H}_K$ iff $\sum_k c_k^2 k^{\alpha} < \infty$. Larger $\alpha$ imposes a harsher penalty on high-frequency coefficients, shrinking the function space: $\alpha_S > \alpha_T \Rightarrow \mathcal{H}_S \subset \mathcal{H}_T$. Thus a "stronger'' model possesses a richer representational space that includes all functions the weaker model can express plus additional rough, high-frequency components.
>
> W5: Sanity checks on effective noise and heavy-tailed task.
>
> A5: Empirical Validation of Effective Noise: In the SGD training, gradient noise arises not only from data sampling but also from the irreducible error inherent in labels, and distillation reduces this bias by replacing hard labels with soft teacher predictions. Our results track the [variance of  mini-batch gradient](https://i.postimg.cc/kXMLBKVh/gradient-variance-comparison.png) (link):
> $\mathrm{tr}(\mathbb{E}[(\hat{\nabla} L_{\mathrm{batch}}-{\nabla} L_{\mathrm{full}})(\hat{\nabla} L_{\mathrm{batch}}-{\nabla} L_{\mathrm{full}})^\top]).$
>
> Empirical Validation of Heavy-tail: We report the observed [heavy-tailed](https://i.postimg.cc/SNBVqddL/heavy-tail.png) (link) phenomenon of the projections onto eigenvectors and the exponentials ($\beta$).
> |Model Name|GPT-2 Base|GPT-2 Medium|GPT-2 Large|DeiT-Base|
> |----------|----------|------------|-----------|---------|
> |$\beta$|1.45|0.04|0.01|0.09|
>
> W6: $N,n$ guidance for estimating DER, choosing early stopping, and sensitivity to $\delta$.
>
> A6: We wish to clarify that our definitions of DER and PGR are non-asymptotic and directly computable as the ratio of test errors. For choosing $n$ in KD, one simply trains the student to convergence so that it fully inherits the teacher's learned representation. For choosing $n$ in W2S, Theorem 5 gives an explicit prescription $n^\star = \tilde{\mathcal{O}}\big((k^\dagger)^{2\alpha_S/(2\alpha_S+1)} \cdot N^{1/[\alpha_T(2\alpha_S+1)]}\big)$. Regarding $\delta$: it is not a hyperparameter but an emergent quantity $\delta(k^\star)\approx 2n\gamma\lambda_{k^\star}$ determined by the SGD dynamics.

---

> > ### Author Rebuttal · Reviewer_rAs7 · 2026-04-02
> >
> > Thank you for the detailed rebutal. I appreciate the effort you put into clarifying the theoretical nuances and providing additional empirical context.
> >
> > Here is my feedback based on your responses:
> >
> > * **Alignment Bias Proxy (W2):** The new empirical probe on ImageNet comparing the fit of the teacher's output versus the residual is a practical and convincing way to validate the geometric consistency assumption. Please ensure this experiment and its methodology are explicitly detailed in the final manuscript or appendix, as it greatly bridges the gap between theory and practice.
> > * **Inconsistent Definitions of $k^*$ (W3.1):** I understand that the varying definitions of the effective dimension $k^*$ (e.g., the different log factors and thresholds) are intentional artifacts of the distinct proof techniques required for different lemmas. However, it is critical that you explicitly state this in the main text when the notation shifts, so readers are not left confused.
> > * **Intuition on Model "Strength" (W4):** Your explanation connecting a larger $\alpha$ to faster eigenvalue decay, fewer learned features at a given $N$, and a harsher penalty on high-frequency coefficients perfectly clears up the intuition. Adding this exact explanation early in the paper will be highly beneficial for readers.
> > * **Empirical Sanity Checks (W5 & W6):** The newly provided data regarding mini-batch gradient variance and the heavy-tailed projections for models like GPT-2 and DeiT are welcome additions. They help ground the theoretical claims in observable deep learning phenomena.
> >
> > **Final Verdict:**
> > Your rebuttal successfuly clarified my primary confusions regarding the notation, the intuition behind the spectral parameters, and the practical measurability of your assumptions. However, the core contribution remains a heavily theoretical analysis bound to a specific linear/spectral framework. While this is technically solid and insightful, the scope inherently limits its immediate broader impact.
> >
> > Therefore, I believe my initial assessment accurately reflects the paper's clear merits alongside its scoping limitations. I will maintain my score.

---

> > > ### Author Response · Authors · 2026-04-06
> > >
> > > Thank you very much for your thoughtful response and for acknowledging that our rebuttal successfully resolved your primary confusions. We are glad that the new empirical probes, the intuition for model "strength", and the clarifications on notations were convincing and helpful.
> > >
> > > We respect your final verdict regarding the inherent scope of our linear/spectral framework. We truly appreciate your recognition of the technical solidity of our analysis and thank you again for your time and constructive feedback.

---

### Official Review · Reviewer_F2si · 2026-03-06

**Soundness:** 3
**Presentation:** 3
**Significance:** 3
**Originality:** 3
**Overall Recommendation:** 4
**Confidence:** 4

**Summary:**

This paper investigates the problem of knowledge transfer within the context of high-dimensional linear regression, providing a unified theoretical framework for understanding knowledge distillation, self-distillation, and weak-to-strong generalization. For each specific regime, the authors derive distinct theoretical insights and characterizations. Furthermore, experiments are conducted to validate their theoretical findings across these different settings.

**Compliance With Llm Reviewing Policy:**

Affirmed.

**Final Justification:**

I maintain a positive view of this manuscript, and my evaluation remains unchanged.

**Key Questions For Authors:**

See weakness.

**Limitations:**

The main limitation is that the theoretical analysis is confined to high-dimensional linear regression models.

**Strengths And Weaknesses:**

**Strength**

1. The paper is well-structured and the writing is clear.

2. The authors provide a rigorous and complete theoretical perspective of knowledge transfer within the high-dimensional linear regression setting.

3. The theoretical findings are well-supported by experimental results.

**Weakness**

1. I believe the paper meets the acceptance threshold, although I did not rigorously check every proof. The primary weakness of this work is that the theoretical analysis is strictly confined to high-dimensional linear regression. While this setting allows for rigorous proofs, it may not fully capture the complex, non-linear dynamics, feature learning, and hierarchical representations inherent in deep neural networks. The transition from linear intuition to non-linear reality remains a significant open question that is not addressed here. The authors should discuss the robustness of the theory. When extending it to more complex scenarios, they should predict what new phenomena may arise or what theoretical difficulties may be encountered.

2. The Assumption 4 in w2s generalization seems too strong. You assume that  teacher and student are structurally sufficient over the downstream task. This simplifies the problem too much, as the real challenge of w2s generalization is how a student learns features that the teacher cannot represent at all.

---

> ### Author Rebuttal · Authors · 2026-03-31
>
> We sincerely thank the reviewer for their positive feedback.
>
> **W1: Reliance on the linear regression assumption vs. modern deep learning.**
> We completely agree that modern deep neural networks are highly non-linear. However, our choice of high-dimensional linear regression is deliberate and well-motivated as a rigorous proxy for the **downstream fine-tuning regim**, which is exactly where KD, W2S, and Self-Distillation are deployed in practice.
>
> **(a) Why linear models are the right proxy for fine-tuning.**
> As rigorously demonstrated by Malladi et al. (2023), fine-tuning pre-trained models operates in a localized, lazy-training regime where the optimization dynamics are mathematically well-approximated by kernel regression—i.e., linear regression in the RKHS. Our spectral framework thus directly models the implicit regularization of SGD during practical fine-tuning.
>
> **(b) What changes in the non-linear (feature-learning) regime.**
> We appreciate the reviewer's suggestion to discuss extensions. The central theoretical difficulty in the non-linear regime is that *the feature map itself evolves during training*, making the effective covariance spectrum non-stationary. This introduces two new phenomena absent in our framework: (1)**Adaptive spectral structure**: SGD not only traverses a fixed spectrum but actively reshapes it, coupling feature learning with optimization dynamics; (2)**Representation alignment**: the teacher and student may develop different feature bases during training, breaking our spectral compatibility assumption. Rigorously analyzing these coupled dynamics is an exciting open direction.
>
> **(c) Precedent for linear proxies.**
> We note that the linear regime has proven remarkably predictive of deep learning phenomena. Recent works have shown that linear models accurately explain neural scaling laws (Atanasov et al., 2024; Lin et al., 2025), double descent, and benign overfitting—all phenomena first observed in deep networks. Our spectral mechanisms (Horizon Expansion and $\delta$-Damping) similarly capture the core optimization logic that drives KT in practice.
>
> We will add an explicit discussion of these non-linear extensions and their theoretical challenges in the revised paper.
>
> **W2: Assumption 4 (Sufficient Expressivity) is too strong.**
> We thank the reviewer for this important point. Our response has two parts.
>
> **(a) The assumption is natural for the fine-tuning regime.**
> Assumption 4 ($\Pi_T = \Pi_S = I_D$) states that both models can represent the downstream task. This is well-justified in the fine-tuning setting where W2S is deployed: both teacher and student start from *pre-trained* representations, and downstream tasks have **low intrinsic dimensionality** (Aghajanyan et al., 2020). Even a weak teacher's pre-trained features span the relevant task subspace—the teacher's limitation is not representational but *statistical* (insufficient data or optimization noise). This is precisely the regime where W2S succeeds empirically (Burns et al., 2023).
>
> **(b) The assumption does not trivialize the problem.**
> Even under structural sufficiency, the W2S phenomenon remains highly non-trivial: it is *a priori* unclear why a student trained on a teacher's noisy labels should outperform the teacher itself. Our theory reveals the precise mechanism—Spectral Denoising via $\delta$-damping—and shows this requires careful early stopping (the student *degrades* if trained to convergence). Furthermore, our theory **predicts failure cases**: when the intrinsic dimension $k^\dagger$ is large relative to the teacher's effective horizon, the denoising benefit vanishes. This aligns with empirical observations that W2S struggles on highly complex tasks.
>
> Regarding the reviewer's point that the ``real challenge is how a student learns features the teacher cannot represent'': we respectfully note that this scenario—where the teacher is *representationally* insufficient—falls outside the standard W2S fine-tuning paradigm and into a fundamentally different regime (closer to semi-supervised learning from corrupted labels). Analyzing this complementary regime is an interesting direction; however, it requires entirely different tools (e.g., feature learning theory) and we believe the fine-tuning regime already captures the dominant practical use case.

---

> > ### Author Rebuttal · Reviewer_F2si · 2026-04-01
> >
> > Thank you for youre response. I maintain my score.

---

> > > ### Author Response · Authors · 2026-04-06
> > >
> > > Thank you for reviewing our rebuttal and for acknowledging that our response has resolved your concerns. We truly appreciate your time, effort, and the valuable feedback you provided during the review process.

---

### Official Review · Reviewer_AiwL · 2026-03-12

**Soundness:** 1
**Presentation:** 2
**Significance:** 2
**Originality:** 3
**Overall Recommendation:** 4
**Confidence:** 2

**Summary:**

This work investigates high-dimensional linear regression with online SGD within a teacher-student training framework, which recovers knowledge distillation, self-distillation, and weak-to-strong generalization. The authors unveil distinct mechanisms for knowledge distillation and weak-to-strong generalization, namely spectral horizon expansion and spectral denoising.

**Compliance With Llm Reviewing Policy:**

Affirmed.

**Final Justification:**

"The authors' response addressed my questions and concerns. In particular, the response on how their assumptions for knowledge distillation and weak-to-strong generalization are related to practical scenarios resolved my main concerns. Therefore, I lean toward acceptance.

**Key Questions For Authors:**

- Could the authors provide more intuitive explanations behind the spectrum cutoffs $k^\*_\nu$ and $k^\*$ described in Lemma 1 and Theorem 1? Currently, it is difficult to grasp why the learning progress is specifically characterized by these cutoff values.
- Could the authors explain why $k_\nu^\*$ and $k^\*$ have different dependencies on the number of data ($n,N$)?
- Could the authors provide a concrete example of models satisfying all technical assumptions for both knowledge distillation and weak-to-strong generalization together? Additionally, could they explain how these examples could be interpreted as weak and strong models in a natural way?

**Limitations:**

This work does not discuss its limitations and potential societal impact. Although the work is primarily theoretical, I suggest the authors explicitly address the limitations of their theoretical settings.

**Strengths And Weaknesses:**

The paper provides a rigorous theoretical analysis of a novel teacher-student training setting in linear regression, identifying two key mechanisms: spectral horizon expansion and spectral denoising. However, I have significant concerns on problem setting and technical assumptions considered in the paper. However, I have significant concerns regarding the problem setting and technical assumptions considered in the paper. While the authors argue that they analyze knowledge distillation and weak to strong generalization which have contrasting teacher student capability gaps within a unified framework, it appears that these two mechanisms are derived from entirely different scenarios rather than representing contrastive aspects of a unified framework. Specifically, I observed that switching the roles of the teacher and student in one case does not lead to the emergence of the other. In real-world scenarios, KD and W2S generalization are often viewed as symmetric, simply swapping a strong and weak model should transition one setting into the other. Furthermore, while the authors provide some justification (Remark 3) for the expressive capability gap in the KD setting (Assumption 2), the intuition behind the weak-to-strong generalization case is less clear. I found it difficult to grasp how Assumptions 4 and 5 fundamentally lead to the weak teacher, strong student case. A more detailed discussion on why these specific assumptions represent the relative strengths of the models is necessary.

In addition, I believe the presentation of the paper could be further improved, and here are some suggestions:

- There are some abuse of notation which might confuse potential readers. For instance, in Algorithm 1, the learning rate scheduling function $S(t)$ is applied identically to both teacher training and student training. However, these should be represented as distinct functions because the number of training data points differs in each case. Additionally, the notation for the direct learning case is not well established. It remains unclear because the symbol $\nu$ is used to represent both the student and the teacher, yet for the iteration count, it only uses $N$, which specifically denotes the training iteration of the teacher model. This makes it confusing whether the result is intended to apply only to the teacher model or to both models.
- It is difficult to check the technical details in the Appendix because the main text lacks direct reference links or pointers to the corresponding proofs near the main results.
- I expected a lower bound result on direct learning to establish a formal comparison with teacher student training, but I could not find such a result in the main text. However, I noticed that a lower bound for the direct learning case is provided in the Appendix (e.g., Theorem 7). I encourage the authors to include direct links or references to this result within the main text; adding further discussions on these error bounds would lead to a better understanding of the overall context.
- (Minor) The current draft numbers every displayed equation block, even when it is never referenced later. Using unnumbered equations in those cases would avoid unnecessary clutter.

---

> ### Author Rebuttal · Authors · 2026-03-31
>
> Q1: Intuitive explanation of the spectrum cutoffs ($k^\star$).
>
> A1: Taking the 1D expected bias as a simplified example, $(w_N - w_\star) \approx (1 - \gamma \lambda)^N (w_0 - w_\star) \approx e^{-\gamma \lambda N} (w_0 - w_\star)$. This reveals a sharp phase transition at $\lambda \gamma N \approx 1$. For dimensions where $\lambda \gamma N = \omega(1)$, the error decays exponentially (the feature is learned); where $\lambda \gamma N = o(1)$, the parameter remains stuck at initialization (the feature is unlearned). With eigenvalues decay ($\lambda_1 \ge \dots \ge \lambda_d$), SGD learns these dimensions sequentially. At any step $N$, $k^\star$ sharply separates the learned head ($\lambda_k \gamma N > 1$) from the unlearned tail ($\lambda_k \gamma N < 1$). Therefore, $k^\star$ quantifies the learning progress, shifting rightward to capture higher-frequency information as training proceeds.
>
> Q2: Different dependencies on data size for $k^\star_\nu$ and $k^\star$.
>
> A2: We clarify that $k^\star_\nu$ and $k^\star$ share the identical physical nature (see A1) while serving distinct technical purposes in our bounds. Specifically, in Lemma 1, the definition of $k^\star_\nu$ incorporates a $2\ln N$ factor to ensure an $\mathcal{O}(1/N^2)$ convergence rate for the bias within the learned head. On the other hand, in Theorem 1, $k^\star$ is formulated using the $\delta/4$ threshold. This cutoff is fundamentally necessary to delineate the boundary where the $\delta$-damping effect emerges.
>
> Q3: Concrete example of models satisfying all technical assumptions.
>
> A3: The KD and W2S assumptions characterizes different regimes and are not intended to hold simultaneously. For KD, consider NTKs. Standard NTKs on the hypersphere $\mathbb{S}^{d-1}$ satisfy Assumption 1, and their shared spherical harmonic eigenfunctions guarantee feature compatibility (Definition 1).
> Relative model capability (Assumptions 2) is rigorously captured by spectral decay rates. For example, a ReLU network exhibits slow polynomial decay $\mathcal{O}(k^{-\alpha})$, whereas a Sigmoid network exhibits exponential decay $\mathcal{O}(e^{-\beta k})$. In our framework, ReLU is stronger because it has slower decay. This perfectly mirrors empirical practice: ReLU consistently outperforms Sigmoid because its slower spectral decay allows it to capture sharp, high-frequency features. For W2S, see A6.
>
> W1: The two mechanisms (KD W2S) are derived from entirely different scenarios rather than a unified framework.
>
> A4: Our framework is unified not merely superficially, but at the deep algorithmic level of **the implicit regularization of SGD**  (characterized by $k^\star$) across distinct problem settings and data spectra.
> The $k^\star$ engine drives both phenomena:In KD, the student leverages the $k^\star$ progression to expand its learned spectrum. In W2S, the student relies on the same $k^\star$ boundary to halt learning before absorbing high-frequency noise ($\delta$-damping). Therefore, while the scenarios contrast, their underlying engine is identical. We provide some [empirical evidence](https://i.postimg.cc/wM3ZR8xK/early-stopping.png) (link).
>
>
> W2:  Switching the roles of the teacher and student in one case does not lead to emergence of the other. In real-world scenarios, KD and W2S are often viewed as symmetric.
>
> A5: We respectfully clarify that in practice, KD and W2S are **fundamentally asymmetric**.
> This asymmetry stems from their different optimization dynamics. In standard KD, the student is trained to *convergence* to fully replicate the teacher's capabilities. In contrast, W2S strictly forbids convergence; perfectly mimicking a weak teacher inevitably causes performance collapse. As established in the original W2S work by OpenAI (Burns et al. 2023), W2S intrinsically relies on *early stopping* to capture a transient phase. Our distinct theoretical settings deliberately formalize these divergent optimization realities.
>
> W3: How Assumptions 4, 5 fundamentally lead to the w2s case.
>
> A6: These assumptions focus more on characterizing a strong student than the relative power of the two. Even if the teacher is stronger, a sufficiently expressive student can still leverage its excess capacity to perform spectral denoising. This theoretical mechanism is perfectly applicable to KD on downstream tasks, where PGR is equivalent to our DER (i.e., 1-PGR$\approx$DER).
>
> W4: There are some abuse of notation in direct learning.
>
> A7: The use of $\nu$ and $N$ in direct learning is not an abuse of notation. Recall that in the definition of DER, we compare the student trained with teacher labels ($\mathcal{E}_{T2S}(N,n)$) with the student directly trained with $N$ raw labels ($\mathcal{E}_S(N)$) to see whether it is necessary and efficient to apply the distillation algorithm under the same labeled data budget $N$.
>
> W5: Missing Appendix Links:
> Will add reference links in revision.
>
> W6: Lower Bound Results:
> Due to 8-page limit, we put lower bound results to appendix.

---

> > ### Author Rebuttal · Reviewer_AiwL · 2026-04-04
> >
> > Thanks to the authors for their detailed rebuttal. Some of my concerns are resolved, but I have a follow-up question regarding the symmetry of knowledge distillation (KD) and weak-to-strong generalization (W2S). While I acknowledge that KD and W2S have asymmetric optimization, as the authors mentioned, to my knowledge, they still have symmetry in their architecture. Therefore, my intent in weakness 2 was considering switching the architectures (the assumptions on the teacher and student models), not the whole structure. In this context, could the authors provide a further response to this?

---

> > > ### Author Response · Authors · 2026-04-06
> > >
> > > **Regarding Switching KD and W2S**
> > >
> > > We appreciate this insightful follow-up. A better perspective to view our framework is through the actual phases of modern deep learning: our KD section models **from-scratch pre-training**, while the W2S section models **downstream fine-tuning**. This physical distinction fundamentally breaks the symmetry:
> > >
> > > **1. The Pre-training Regime (KD, Assumptions 2 & 3):**
> > > If we attempt to swap roles during from-scratch training—forcing a strong model to train on a weak model's logits until convergence—the strong model will inevitably bottleneck and **collapse to the weak model's capacity**. From-scratch training simply cannot elicit better performance from a weak supervisor. Because symmetry is physically broken here, we **only discuss KD** under the pre-training scope.
> > >
> > > **2. The Fine-tuning Regime (W2S, Assumptions 4 & 5):**
> > > Conversely, $\delta$-damping is a fine-tuning phenomenon. While this denoising effect technically exists in any very expressive model (A6), Assumption 4 implies the "geometrically consistent" condition formalized in **Lemma 2**.
> > > If the teacher is vastly stronger than the student, Lemma 2 mathematically lower-bounds the student's minimum achievable loss strictly *above* the teacher's loss. In this regime, the student simply inherits the teacher's superior representations; this massive **inheritance effect completely overshadows the $\delta$-damping mechanism**.
> > >
> > > **Conclusion:**
> > > Therefore, $\delta$-damping only emerges as the *dominant, observable* driver of performance when the model can actually surpass its supervisor's loss. This only happens when the student is not weaker than the teacher (W2S/Self-distillation), or when both are exceptionally strong. Ultimately, our framework is mathematically unified under the same SGD spectral dynamics; however, the dominating terms (Horizon Expansion vs. $\delta$-damping) naturally shift depending on the training phase and the capability gap dictated by Lemma 2.
> > >
> > > **Experiments on the other regime where $\delta$-damping effect happens:**
> > >
> > > We conduct knowledge distillation experiments on the ImageNet dataset. The training set is subsampled to 80,000 images, while the validation set is halved to 25,000 images for evaluation.
> > >
> > > We select two models with strong representational capacity, one slightly weaker than the other. The teacher model (ViT-L/16) and the student model (ViT-B/16) are both pre-trained on ImageNet. We train with linear probing (only the classification head is trained) under the classical knowledge distillation setting.
> > >
> > > We employ a temperature T=4.0 and a mixing coefficient α=0.1 for the cross-entropy loss, while (1-α) is used for the KD loss. We record the [test loss and accuracy](https://i.postimg.cc/bwCRcmM4/distillation.png) (link) after every epoch.
> > >
> > > | Epoch | 0 | 1 | 2 | 3 | 4 | 5 | 6 | 7 | 8 | 9 | 10 | 11 | 12 | 13 | 14 | 15 | 16 | 17 | 18 | 19 | 20 |
> > > | :------- | :----- | :----- | :----- | :----- | :----- | :----- | :----- | :----- | :----- | :----- | :----- | :----- | :----- | :----- | :----- | :----- | :----- | :----- | :----- | :----- | :----- |
> > > | Loss     | 6.9660 | 6.5126 | 6.2643 | 5.9205 | 5.7324 | 5.4514 | 5.2286 | 5.0151 | 4.8918 | 4.7195 | 4.6188 | 4.5403 | 4.4244 | 4.4036 | 4.4155 | 4.4284 | 4.4331 | 4.4540 | 4.4753 | 4.4812 | 4.4873 |
> > > | Accuracy | 0.0013 | 0.0130 | 0.0866 | 0.2701 | 0.4676 | 0.5984 | 0.6681 | 0.7062 | 0.7282 | 0.7420 | 0.7509 | 0.7573 | 0.7589 | 0.7608 | 0.7596 | 0.7588 | 0.7576 | 0.7565 | 0.7512 | 0.7502 | 0.7498 |

---

### Official Review · Reviewer_gorP · 2026-03-24

**Soundness:** 3
**Presentation:** 2
**Significance:** 2
**Originality:** 3
**Overall Recommendation:** 4
**Confidence:** 4

**Summary:**

The paper studies the dynamics of stochastic gradient descent (SGD) with step-decay learning rates in high-dimensional linear regression. Building on this setup, it develops a unified theoretical framework to analyze both knowledge distillation and weak-to-strong generalization.

The authors introduce a teacher–student setting with potentially different feature representations and formalize knowledge transfer as a two-phase SGD procedure: first training the teacher on labeled data, and then training the student on teacher-generated labels.

In the knowledge distillation regime, the analysis shows that the teacher’s outputs provide a better-aligned and lower-noise target for the student, enabling improved sample efficiency and faster convergence under appropriate spectral conditions.
In the weak-to-strong regime, the authors demonstrate that a higher-capacity student can outperform a weaker teacher by leveraging spectral filtering: the student suppresses high-frequency noise in the teacher’s predictions while recovering the low-dimensional structure of the ground truth.

The paper concludes with synthetic and real-world experiments that provide empirical support for the proposed theoretical mechanisms.

**Compliance With Llm Reviewing Policy:**

Affirmed.

**Final Justification:**

The authors rebuttal addressed some concerns regarding the assumptions and I adapted my score accordingly

**Key Questions For Authors:**

1. **Learning rate schedule.**
   The analysis relies on a step-decay schedule. How would the results change under iterate averaging or tail averaging, which are known to achieve optimal rates? In particular, do the spectral cutoff $k^*$ and the $\delta$-dependent damping effects persist?

2. **Kurtosis assumption.**
   The fourth-moment assumption may be restrictive. For instance, for one-hot each with prob $p_i$, the kurtosis can scale as $1/p\_{\min}$ and become large when $(p\_{\min}$ is small. How sensitive are the results to this assumption ?

3. **Spectral compatibility.**
   The assumption that student features are a linear transformation of teacher features is quite strong. How is it rigorous in the NTK limit? Can the authors provide further empirical evidence or robustness checks? Similarly in the next assumption, where the teacher dominates in every eigendirection.

4. **Self-distillation setup.**
   In practice, self-distillation often initializes the student from the trained teacher, whereas the analysis assumes fresh initialization. How would this affect the results ?

---

Minor Comments

5. **Terminology.**
   The paper introduces many new terms and mechanisms, which hurts readability. Simplifying or consolidating terminology would help.

6. **Notation consistency.**
   In Lemma 1, notation appears inconsistent (e.g., $\Sigma$ vs $\Sigma_\nu$, $\Pi$ vs $\Pi_\nu$ ).

**Limitations:**

yes

**Strengths And Weaknesses:**

**Strengths**
- Theoretical understanding of knowledge transfer remains an important open problem, and this work makes meaningful progress toward a unified explanation across distillation and weak-to-strong generalization.
- The teacher–student framework with distinct feature mappings is a strong contribution, enabling precise characterization of representation mismatch and transfer dynamics.
- The paper identifies clear spectral mechanisms—horizon expansion and denoising—that provide intuitive and theoretically grounded explanations for the effectiveness of knowledge transfer.

**Weaknesses**
- The analysis is limited to fixed feature representations and does not capture feature learning, which is central to modern deep neural networks.
- The results rely on restrictive assumptions (e.g., bounded kurtosis,  spectral compatibility, uniform teacher superiority, and low intrinsic dimensionality) that are not sufficiently validated empirically.
- The connection between the linear regression setting and practical deep learning models remains somewhat indirect.

---

> ### Author Rebuttal · Authors · 2026-03-31
>
> Q1: Learning rate schedule.
>
> A1: Yes, the spectral cutoff $k^\star$ and $\delta$-damping both persist under averaging. For iterates averaged from step $s$ to $s{+}n$, the spectrum naturally partitions via two thresholds: $k_1^\star=\max\{j:\lambda_j\ge 1/(\gamma n)\}$ and $k_2^\star=\max\{j:\lambda_j\ge 1/(\gamma(s{+}n))\}$. The bias evaluates at $k_1^\star$ (analogous to $k_\nu^\star$ in our paper), and the variance decomposes as
>
> $\text{var} \leq C\sigma^2 \left( \frac{k^\star_1}{n} + 4\eta \sum_{k^\star_1 < j \leq k^\star_2} \lambda_j + 16\eta^2(s+n) \sum_{j \geq k^\star_2} \lambda_j \right).$
>
> The averaged damping factor is $\bar\delta=\frac{1}{n}\sum_{t=s+1}^{s+n}\delta_t\approx 2(s+\frac{n+1}{2})\gamma\lambda_{k^\star}$, giving $\bar\delta\approx(n{+}1)\gamma\lambda_{k^\star}$ for uniform averaging ($s{=}0$) and $\bar\delta\approx(3n{+}1)\gamma\lambda_{k^\star}$ for tail averaging ($s{=}n$).
> We note that while iterate averaging achieves optimal rates for many targets, it is suboptimal for highly smooth functions due to bias saturation (Dieuleveut \& Bach, 2016), so step-decay is not merely a convenience.
>
> Q2: Kurtosis assumption.
>
> A2: One-hot vectors satisfy the weaker condition $E[xx^Txx^T]\preceq R^2E[xx^T]$ with R = 1. Our proofs use the fourth-moment condition in two places: (i)Eq. 76$\to$77, where the weaker condition applies directly; (ii) Lemma 3 (Eq. 87), which for one-hot vectors simplifies to  $E[\phi\phi^\top\eta_0^{\otimes 2}\phi\phi^\top] = diag(p_i(\eta_0^{\otimes 2})_{ii})\preceq diag(p_i)\Vert \eta_0 \Vert^2$,
>
> so  $\psi\Vert w_0-w_\star\Vert_\Sigma^2$ is replaced by $\Vert w_0-w_\star\Vert_2^2$  with analogous results. In the W2S regime this is always bounded since $w^\star$ is finite-dimensional. In KD scenario, $\Vert w_0-w_\star\Vert_2^2$ can be unbounded, making results sensitive to $\psi=1/p_{\min}$.
>
> Empirical validation: We compute global kurtosis $\psi=\mathbb{E}[\|\boldsymbol\phi\|^4]/(\mathbb{E}[\|\boldsymbol\phi\|^2])^2$ and directional kurtosis $\psi_\mathbf{u}=\mathbb{E}[(\mathbf{u}^\top\boldsymbol\phi)^4]/(\mathbf{u}^\top\Sigma\mathbf{u})^2$ (max over 1000 random $\mathbf{u}$) on ResNet-50 features:
>
> | Dataset   | $\psi$ | $\psi_{\mathbf{u}}(\max)$ |
> | --------- | ------ | ------------------------- |
> | CIFAR-100 | 1.122  | 5.394|
> | ImageNet  | 1.061  | 4.777|
> | UTKFace   | 1.031  | 6.236|
>
> All values are moderate, confirming the assumption is mild for pretrained features.
>
> Q3: Spectral compatibility.
>
> A3: In the NTK limit on $\mathbb{S}^{d-1}$, the eigenfunctions are spherical harmonics shared by all architectures; different NTKs vary only in their eigenvalues, making them automatically compatible. We stress that our framework does \emph{not} require component-wise domination ($\lambda_{k,T}\ge\lambda_{k,S}$ for all $k$); we only require the macroscopic decay rates to satisfy $\alpha_T\le\alpha_S$. Because our framework permits arbitrary basis rotations ($Q_{\mathrm{Trans}}$), the models need not be aligned dimension-by-dimension for the transfer mechanisms to hold.
>
> Empirical validation: For empirical verifications of alignment bias (as proxy to compatibility)，see A2 in rebuttal to rAs7.
>
> For decay rates of eigenvalues see [our result](https://i.postimg.cc/QdBzXVc9/eigen-decay.png) (link) .
>
> Q4: Self-distillation setup.
>
> A4: If the student were initialized with the exact teacher weights and trained on teacher's soft labels directly, the transfer risk would trivially be zero. The gradient would be zero, and no learning. In the empirical reality, such training employs data augmentation, dropout or joint objectives, which are not considered in this theoretical paper.
>
> Q5 \& Q6: Terminology and Notation consistency: We sincerely appreciate your careful reading and feedback. Will fix in revision.
>
> W1 \& W3: Feature Learning \& Practical Deep Learning.
>
> A5: Our core thesis is that the successes of KT are driven by the \emph{implicit regularization of SGD} (characterized by $k^\star$) across different problem settings and data spectra, not by feature learning per se.  In practice, W2S and KD are deployed during downstream fine-tuning, where feature updates are minimal and dynamics are rigorously approximated by kernel regression (Malladi et al., 2023). Since our linear regression framework is mathematically equivalent to RKHS optimization, our spectral insights directly explain the internal logic of fine-tuning practical networks.
>
> W2: Assumption Validations.
>
> A6: For bounded kurtosis see A2. For spectral compatibility and teacher superiority see A3. For low intrinsic dimensionality, we have verified the following (beyond figure 3.c. in our paper). [Details](https://i.postimg.cc/7hk9t9mX/intrinsic-dimensions.png) .
>
> |Name|Baseline Acc.|k_{90\%}|
> |-----|-----|----|
> |resnet50_dino|64.1%|116|
> |vits16_dino|65.5%|66|
> |vitb16_dino|69.0%|63|
> |vits8_dino|69.8%|49|
> |vitb8_dino|72.3%|63|
> |regnety_16gf|73.9%|37|
> |efficientnet_b4 |77.6%|30|
> |swin_t|78.9%|59|
> |convnext_base|83.2%|24|

---

> > ### Author Rebuttal · Reviewer_gorP · 2026-04-04
> >
> > I thank the authors for their reply, I adjusted the score accordingly.

---

> > > ### Author Response · Authors · 2026-04-06
> > >
> > > We sincerely thank the reviewer for taking the time to read our rebuttal and for adjusting the score. We are very glad that our response has adequately addressed your concerns. We deeply appreciate your constructive feedback.

---

### Decision · Program_Chairs · 2026-04-30

**Decision:**

Accept (regular)

**Comment:**

This paper presents a unified theoretical framework to explain the efficacy of Teacher-Student Knowledge Transfer (KT) across disparate regimes, specifically Knowledge Distillation (KD) and Weak-to-Strong (W2S) generalization. By utilizing a spectral analysis of online Stochastic Gradient Descent (SGD) dynamics within high-dimensional linear regression, the authors identify two distinct drivers of transfer efficiency. They characterize Spectral Horizon Expansion as the mechanism behind KD, allowing a student to capture statistically inaccessible high-frequency signals, and Spectral Denoising as the engine for W2S and self-distillation, where a strong student leverages its capacity to filter out a weak teacher's optimization noise.

The initial reception of the paper was uniformly positive, with all four reviewers recommending a Weak Accept. Reviewers praised the paper's rigorous risk decomposition, the intuitive formalization of spectral mechanisms, and the clear separation of optimization error from alignment bias. However, a universal concern among the reviewers was the paper's confinement to a linear regression framework, which they felt might not fully capture the complex, non-linear feature learning inherent in modern deep neural networks. Additionally, reviewers raised questions regarding the practicality of several theoretical assumptions (e.g., bounded kurtosis, sufficient expressivity, and spectral compatibility), noted some notation inconsistencies, and requested a practical method to measure alignment bias empirically.

During the rebuttal, the authors provided a strong defense of their framework. They effectively addressed the linear scope limitation by connecting it to the downstream fine-tuning regime, noting that fine-tuning dynamics are well-approximated by kernel regression. The authors also provided new empirical validations, including an ImageNet proxy for measuring alignment bias, and clarified that their kurtosis assumptions are empirically mild for pretrained features. Furthermore, they successfully resolved Reviewer AiwL's concerns by clarifying the optimization asymmetry between KD (viewed as pre-training) and W2S (viewed as fine-tuning). Following the discussion, Reviewers gorP and AiwL stated their main concerns were resolved and adjusted their perspectives positively, while Reviewers F2si and rAs7 maintained their positive scores, acknowledging the paper's technical solidity despite the inherent boundaries of a linear framework.

Overall, the reviews reflect a clear consensus that this paper is technically sound and advances the theoretical understanding of knowledge transfer. While the reviewers correctly identify that the linear regime has scope limitations regarding deep non-linear networks, the authors successfully argued that their spectral insights still directly explain the internal logic of fine-tuning practical networks. Given the theoretical contributions, the unification of KD and W2S under a single spectral framework, and the authors' comprehensive rebuttal, the paper warrants inclusion in the conference. My final recommendation is Accept